# 3D Printing Application in Wood Furniture Components Assembling

**DOI:** 10.3390/ma15082907

**Published:** 2022-04-15

**Authors:** Antoniu Nicolau, Mihai Alin Pop, Camelia Coșereanu

**Affiliations:** 1Faculty of Furniture Design and Wood Engineering, Transilvania University of Brasov, B-dul Eroilor, nr. 29, 500036 Brasov, Romania; antoniu.nicolau@unitbv.ro; 2Faculty of Materials Science and Engineering, Transilvania University of Brasov, B-dul Eroilor, nr. 29, 500036 Brasov, Romania; mihai.pop@unitbv.ro

**Keywords:** additive manufacturing, wood furniture, 3D printed connector, mortise–tenon joint, polylactic acid (PLA)

## Abstract

Additive manufacturing (AM) is used in many fields and is a method used to replace wood components or wood-jointed furniture components in the furniture industry. Replacing wood joints by 3D printed connectors would be an advantage, considering the fact that during the process of assembling furniture, the execution technology of the joints is difficult, time-consuming, and labor-intensive. Advanced technology of AM applied in furniture manufacturing helps the designers to create new concepts of product design, with no limits of shape, number of joints, color, or size. The diversity of 3D printers and AM technologies provides the selection of materials in relation with the applicability of the 3D printed object. In this respect, the objective of the present research is to design a 3D printed connector to be used for jointing three chair components, namely the leg and two stretchers made from larch (*Larix decidua* Mill.) wood, and to use reinforced polylactic acid (PLA) fiberglass (20 wt. %) filament for 3D printing this connector using AM with fused filament fabrication (FFF) technology. The design of the connector, the possibility of using this type of material, and the deposition method of filament were investigated in this research. For this purpose, several evaluation methods were applied: microscopic investigation with 50×, 100×, and 200× magnifications, both of the filament and of the 3D printed connector; mechanical testing of corner joint formed with the help of connector between chair leg and the two stretchers; and a microscopic investigation of the connectors’ defects that occurred after applying the compression and tensile loads on the diagonal direction of the L-type joint. The microscopic investigation of the composite filament revealed the agglomerations of glass fibers into the core matrix and areas where the distribution of the reinforcements was poor. The heterogeneous structure of the filament and the defects highlighted in the 3D printed connectors by the microscopic investigation contributed to the mechanical behavior of L-type connecting joints. The bending moments resulting from compression and tensile tests of the 3D printed connectors were compared to the results recorded after testing, under the same conditions, the normal mortise–tenon joint used to assemble the abovementioned chair components. The larch wood strength influenced the mechanical results and the conclusions of the microscopic investigations, as well as the analysis of the broken connectors after testing recommended the change of connector design and filament deposition direction.

## 1. Introduction

Additive manufacturing (AM) is a constantly expanding field [1] which was first used in the furniture industry in such applications as 3D printing of furniture pieces, components, and joints. The most used 3D printing technologies in furniture manufacturing are fused filament fabrication (FFF), fused deposition modeling (FDM) [2,3], and selective laser sintering (SLS) [2,4], with FFF technology the most accessible. FFF uses thermoplastic materials in the form of filaments, such as acrylonitrile butadiene styrene (ABS), polylactic acid (PLA), polyvinyl alcohol (PVA), polycarbonate (PC), high-density polyethylene (HDPE), or polyethylene terephthalate (PET) [5]. SLS is an expensive technology that uses powders from various materials such as thermoplastic powders (nylon, polyamide, polystyrene, elastomers, and reinforced composites), ceramic powders, or glass powders. SLS technology allows for the printing of large furniture components, or even of an entire piece of furniture [6]. Using parametric design plugins for basic modeling software, interesting Voronoi parameterized microstructures can be obtained [7]. An essential advantage of additive manufacturing using Voronoi’s algorithm is its ability to manufacture complex structures on a small scale. Voronoi microstructures can be understood as metamaterials, for which the material exists on a much smaller scale than the volume it fills along specific directions [7,8,9]. The materials used in FFF printing technology are continuously developed in order to improve their chemical, physical, and mechanical properties, as well as their aesthetics, which have an influence on the appearance of the printed product, and they are usually composite materials [10].

One of the materials used in FFF technology is polylactic acid (PLA). In order to improve its physical and mechanical properties, solutions have been found to reinforce it with various fibers (natural fibers, glass fibers, or carbon fibers) [11,12,13,14]. Depending on the applications taken into account, previous research works were focused on improving PLA’s physical and mechanical properties, such as high gloss and antibacterial properties [11], or tensile mechanical properties of continuous carbon fiber-reinforced 3D printing composites in relation to process parameters [15]. Using wood leachate powder as an additive in PLA-based bio-composites had a positive effect on the physical (surface hydrophobicity, antibacterial properties) and mechanical properties (tensile strength, elongation at break, hardness, and impact strength) compared to the neat PLA sample [16]. A major advantage of using PLA as a printing material is that it is easily biodegradable. PLA is a thermoplastic polyester made from renewable resources such as corn starch, tapioca roots, and sugar cane [17], and it fits to any printer. PLA filaments are the most common type for furniture manufacturing, as well as PLA mixed with wood fibers [2]. Composites of mixed PLA–wood fibers obtained from furniture waste were also investigated as potential materials for 3D printing [13], and recycled PLA filament proved to induce similar mechanical properties to the 3D printed specimens when compared to virgin PLA [18]. PLA was used for printing 3D connectors for frontal parallel joints, which were tested for bending strength [19]. The fractures of the joints occurred in the PLA material of the connectors and the bending strength was lower than in the case of using unjointed spruce wood. PLA reinforced with chopped flax fibers was used to 3D print connectors for bamboo members of a structure, showing potential use, but needs more work to be carried out in order to define the system rigidity [20]. Invisible 3D printed cabinet furniture joints made from ABS proved to have high resistance and stiffness, and potential use for particleboard and medium density fiberboard (MDF) assembling [3].

Fiberglass was found to be used as a reinforcing material in the composition of polyamide-based material for printing an entire piece of furniture with SLS technology [2]. Reinforced PLA fiberglass filaments provide strength and stiffness properties [21], and studies in this field show increases in modulus and tensile strength in these reinforced composites [22].

Two methods are applied in the literature to test the stiffness of joints: diagonal compression and diagonal tensile test [3,23,24,25]. During these tests, loads are applied on L-type joints, both for jointing wood and wood-based panels [26].

The present research has an objective to replace the basic glued mortise–tenon joints used for assembling the chair leg and stretchers with a 3D printed connector in which the three chair components are introduced. This replacement brings the advantage of easing the assembly process and reducing the production time, with operations such as processing and gluing the tenon–mortise joints not being necessary in this case. Considering that glass fiber-reinforced PLA composites develop enhanced mechanical properties [27,28,29,30], reinforced PLA fiberglass (20 wt. %) filament was selected. The FFF technology was used for 3D printing the connectors and a double-nozzle printer CreatBot DX Plus-3D (Henan Creatbot Technology Limited, Zhengzhou City, Henan Province, China). The material and the 3D printing process of the connectors in this research were selected considering the characteristics of the 3D printer (printing temperatures and controlled parameters), the shape of the connector, the biodegradability of the material, the low costs generated by 3D printing including power consumption.

The mechanical strength of the 3D printed connecting joint was investigated on an L-type joint subjected both to diagonal compression and tensile tests. In parallel, the same tests were conducted on mortise–tenon corner joints made from larch (*Larix decidua* Mill.) wood for comparing the results.

## 2. Materials and Methods

### 2.1. Experimental Setup

Usually, the joints used to assemble the chair leg and stretchers are mortise–tenon joints (Figure 1). The experimental study thus investigated the possibility of using 3D printed connection parts between the two chair stretchers and leg.

The software used for 3D modelling of the connector was SolidWorks 3D CAD, version 2016 developed by Dassault Systèmes, France. A SLDPRT-type file was generated and exported for 3D printing to the software of double-nozzle printer CreatBot DX Plus-3D (Henan Creatbot Technology Limited, Zhengzhou City, Henan Province, China). The printer has the following characteristics: usable filament diameter of 2.85 mm, nozzle temperature up to 260 °C, open filament system, maximum printing speed of 200 mm/s, build area of 250 mm × 300 mm × 520 mm, layer height between 0.05 mm and 0.5 mm, resolution of 0.05 mm. The designed connector is presented in Figure 2a. The set position of the connector prepared for printing is shown in Figure 2b. The wooden piece corresponding to the chair leg were introduced in the square section of the connector and the stretchers were introduced in the two rectangular sections of the designed connector. Thus, L-type joints are formed by the three chair components.

The main printing parameters used for 3D printing were: print speed of 50 mm/s; printing temperature of 250 °C; layer height of 0.2 mm; 100% fill density. The 3D printing parameters were selected according to the recommendations made by the filament manufacturer and following the preliminary results obtained by the 3D printed parts in terms of mechanical strength for different printing parameters. The designed connector was intended to replace the normal mortise–tenon joint (Figure 1), commonly used for this purpose. Twelve connectors were 3D printed as samples and used for the construction of L-type corner joints, where larch wood was used for making the wooden elements corresponding to the chair leg and stretchers. Six samples were used for each mechanical test.

### 2.2. Materials

The wooden components (leg and stretchers) were made from larch wood (*Larix decidua* Mill.). Larch wood is a resinous wood with good stability and small moisture movement. It is appreciated for its fine structure, its toughness, and high density compared to other resinous woods. It is strong and sturdy and is often used for yacht building, joinery, stairs, and floors, as well as for external cladding projects and outdoor furniture, due to its good resistance to weathering. In the furniture industry larch wood is mainly used for chairs, benches, and tables and has a large availability on the market at a low cost. The sizes were 210 mm × 35 mm × 19 mm for the chair stretchers and 50 mm × 35 mm × 35 mm for the piece of the leg. The wood moisture content of the larch wood was 9.2%. The tolerances of wooden component sizes were selected for a tight fit with the connector.

Material used in the present research for 3D printing is glass fiber-reinforced PLA filament made with glass fiber (20 wt. %) manufactured by Filaticum, Miskolc, Hungary.

The characteristics of the filament are presented in Table 1.

The complete technical characteristics of the filament are presented in their respective technical data sheets.

The initial filament deposit has the role to create the base for the next layers of deposition and to define a closed contour which will be referred to as perimeter (P) of the 3D printed connector. The layer-by-layer deposition in FFF printing technology is done at an inclination angle of 45° related to the perimeter direction, and perpendicular to the previous layer (Figure 3).

### 2.3. Microscopic Investigation

Microscopic investigation of filament and 3D printed specimens was carried out using an optical microscope from Nikon OmniMet-Buehler (Tokyo, Japan). Three magnifications were used (50×, 100×, and 200×) for optical microscopy of three specimens of reinforced PLA fiberglass filament and three 3D printed samples prepared for mechanical tests. Longitudinal and transversal cuts of the parts and filament were embedded in Dentacryl technical resin for technical uses in mechanical engineering, and subjected to optical microscopy (Figure 4).

Digital microscope INSIZE ISN PM200SB (Insize Co., Ltd., Jiangsu, China) with 10X magnification was used to investigate the defects which occurred after 3D printed connectors were subjected to compression and tensile loads in L-type joints structures. The digital microscope was used with its minimal magnification power and resolution of 1600 × 1200. The objective of microscopy was to better visualize the defects which occurred as a result of failure under tensile and compression loads. Three connectors were investigated before and after compression and tensile tests.

### 2.4. Bending Moment

L-type joints were subjected both to diagonal compression and tensile tests, according to the model provided by other research works [23,24]. The tests were conducted on a Zwick/Roell Z010 universal testing machine, manufactured in Germany. Six samples were used for each test. The testing methods are presented in Figure 5.

The bending moment in the tensile testing (*M_t_*) was calculated according to Equation (1), in N·m.
(1)$$M_{t}=L\cdot \frac{F}{2},$$
where *L* is the moment arm, or the length of the corner joint, in m; *F* is the maximum failure load, in N.

The bending moment in the compression testing (*M_c_*) was calculated according to Equation (2), in N·m.
(2)$$M_{c}=L\cdot F,$$
where *L* is the moment arm, or the length of the corner joint, in m; *F* is the maximum failure load, in N.

Both mortise–tenon L-type joints and 3D printed connecting joints were subjected to compression and tensile testing and the results were afterwards compared.

## 3. Results and Discussion

Microscopic investigation of filaments and 3D printed samples before testing aimed to anticipate the mechanical behavior of corner-joints and to explain the results of the tests. The microscopic investigation of the tested connectors will complete the image and further explain the causes of the samples’ failures.

### 3.1. Microscopic Investigation

#### 3.1.1. PLA–fiberglass Filament

The microscopy of the cross-cut section of filament is shown in Figure 6 with 50× magnification (Figure 6a), 100× magnification (Figure 6b), and 200× magnification (Figure 6c).

In Figure 6a the defects of reinforcement material distribution in the PLA matrix can be observed. The black circular areas in the middle zone are major defects of the filament, which occurred due to glass fiber agglomeration. The 100× magnification (Figure 6b) clearly reveals the major defect of the filament, which is in fact an evulsion of the glass fibers bundle, after crosscutting the sample, proving that the fibers bundle did not have a good adherence to the matrix. In Figure 6c the porosity of the matrix is more visible, as well as the shape, dimensions, and repartition of the glass fibers in the matrix. The PLA porosity was also noticed by SEM micrographs of the PLA’s surface [16]. The first circled area (no. 1) reveals the fact that the matrix did not adhere to the reinforcement. The second circled area (no. 2) shows the gap left by the evulsion of a non-adherent glass fiber to the matrix, in the cross-cutting process. Detail no. 3 represents a common evulsion of the glass fiber in the matrix, which occurred after cutting the sample. The black spots on the microscopic image represent cavities on the surface (evulsions, porosities, cracks). For all three analyzed samples, the central zone of the filament showed an agglomeration of glass fibers (wrong distribution of reinforcement fibers in the matrix).

The microscopy of the longitudinal section of the filament is shown in Figure 7 with 50× magnification (Figure 7a), 100× magnification (Figure 7b), and 200× magnification (Figure 7c). In Figure 7a the presence of glass fiber disposed longitudinally in the right position can be noticed, and shape and length of the glass fibers and evulsions in the composite (i.e., circled area) are visible. The irregular distribution of fiberglass is also visible. In Figure 7b, partial exposure of a glass fiber, continued under a thin film of the matrix (circled area), can be observed. Figure 7c reveals the porosity of the matrix and clearly shows the length and the shape of the glass fibers. The measured lengths of the glass fibers reinforced in the matrix varied from 45 µm up to 715 µm (Figure 7d) for all investigated specimens.

#### 3.1.2. Three-dimensional Printed Connector Samples

The microscopy of the cross-cut section of the 3D printed samples is shown for two samples in Figure 8 and Figure 9, with 50×, 100×, and 200× magnifications. The prints of the deposited layers are visible in Figure 8a (marked area) and magnified in Figure 8b,c. In Figure 8b the triangular defect specific to PLA (circled area) is clearly visible. This defect occurred between two successive layers because of the short sudden dysfunctions of the process in terms of temperature and material fluidity during the deposition. In Figure 8c an end delamination between layers (marked zone) can be seen. In both Figure 8b,c the glass fibers appear with an uneven distribution, and fiberglass bundle can be seen in detail X.

In Figure 9a, the zone marked with 1 shows the perimeter. The next marked area (with 2) highlights two overlaid triangular defects. The area marked with 3 indicates a fiber rupture in the form of fiber–matrix unbound, which is visible as micro-cracking. This defect occurred when the nozzle changed its direction.

The defect marked with 4 in Figure 9a is placed in the perimeter area and represents the end of the perimeter contour, where the nozzle changed the deposition direction of the layer and went in the opposite direction to close the contour on the same line. This defect is detailed more in Figure 9b, where the uneven distribution of reinforcing material into the matrix and glass fiber agglomeration can be also observed.

The microscopy of the longitudinal section for two 3D printed connector specimens are shown in Figure 10 and Figure 11, with 50×, 100×, and 200× magnifications.

In Figure 10a, a delamination on the perimeter is marked on the microscopic image. In Figure 10b, a crack in the perimeter area is highlighted with 100× magnification. A particular defect on the 3D printed part is highlighted in Figure 10c. It is a crack between deposited layer and perimeter. These images also show several defects which occurred because of fiberglass bundle evulsions.

In Figure 11a, three zones are marked on the microscopic image. The areas marked with 1 and 2 show different types of cracks in the matrix, which can be considered macro-defects. The zone marked 3 highlights the incomplete filling of the matrix. When preparing the samples for the microscopic investigation it is possible to reveal two or three layers, because the flatness of the sample is not reached after cutting and sanding the slices. Often, concave surfaces are obtained, as can be seen in Figure 11b,c, where the presence of the two successive deposited layers is highlighted by the orthogonal positions of the reinforcement glass fibers.

### 3.2. Mechanical Performance of Corner Joints

Both mortise–tenon joints and 3D printed connecting joints were subjected to tensile and compression loads in the same conditions of the tests. The calculated values of bending moments for tensile and compression testing of investigated joints are presented in Table 2.

The results presented in Table 2 show that the values recorded for mortise–tenon jointed samples made from larch wood are lower than those obtained by other researchers for the same type of joint made on beech samples with bigger tenon sizes [24]. Instead, the same trend was not observed for the bending moment in compression testing, which was higher than the bending moment in tensile testing, contrary to the results obtained by [24]. Regardless of the type of test (tensile or compression), deep cracks occurred in the structure of the wooden component corresponding to the chair leg. The fractures of these wooden parts were greater in the case of applied tensile load than in the case of compression test (Figure 12). This can explain the lower values of bending moment in the case of the tensile test compared to compression test. The failure mode of the mortise–tenon joint was different for the two tests. In the case of diagonal tensile load, the wood cracks were oriented mostly in the direction of the annual ring contour. Under compression load, the wood cracks occurred in the radial direction of the wood piece. Larch wood has a low mechanical strength compared to beech wood, so the failures occurred in the wood structure rather than in the mortise–tenon joint. The low mechanical resistance of larch wood could explain the lower values recorded for mortise–tenon joints compared to the results recorded by other researchers [23,24]. Apart from the wood species, other factors that have influence on the mechanical resistance include the dimensions of the tenon–mortise joints and the wood moisture content. Newly designed 3D printed fasteners used for corner joints of beech wood, and tested under tension and compression, recorded lower bending moment values, namely 12.34 Nm for tensile and 8.57 Nm for compression tests [26].

The joints made with 3D printed connectors have been tested as shown in Figure 13, for both diagonal compression and tensile loads.

The results from Table 2 show lower bending moment values in the case of compression tests. The failure modes of the 3D printed connectors are different for the two tests. Under compression load, deep cracks occurred on the inside part of the connector (marked with 1 and 2 in Figure 14a), whilst the outside part of the piece was affected by diagonal tensile loads (marked with 1 and 2 in Figure 15a). The areas affected by failure are also highlighted in Figure 13a for compression test and Figure 13b for tensile test. Three-dimensional printed connectors (with and without inner wall) used for the frontal parallel joints [19] and tested for bending load were fractured in the PLA material only for the variant with inner wall, because of the force acting perpendicular to the deposition layers, and forcing the end of the connector to crack. The defects which occurred on the connectors were similar to those presented in Figure 13a.

### 3.3. Microscopic Investigation of the Connectors after Mechanical Testing

The marked areas on the connectors were investigated by 10× magnification to more clearly observe the occurred cracks.

The detail of the area marked with 1 (Figure 14b top) shows delamination between filament layers, due to the fact that the filament layers are oriented parallel to the length of the stretcher; one component of the force acts perpendicular to this direction and tends to detach the layers. Furthermore, on the convex fillet detail from area 2 (Figure 14b bottom), fine cracks occurred between filament layers.

The same delamination between filament layers was also observed on the 3D printed connector after being subjected to tensile test, as seen in the details from Figure 15b; the left one for the cracks which occurred on area 1, and the right one for the cracks which occurred on area 2 marked on the image in Figure 15a.

## 4. Conclusions

The microscopic investigation of the fiberglass-reinforced PLA filament has shown random distribution of glass fiber into the matrix, with areas where the presence of the reinforcement was not detected and areas where fiberglass bundles agglomerated the matrix, especially in the core, where the adherence between these fibers and matrix was poor. These defects could affect the filament strength in certain zones.

On the other hand, the microscopic investigation of the samples cut from 3D printed connectors revealed that defects occurred as a result of printing parameters variation during the printing process. The low mechanical strength of 3D printed connecting joints and delamination between filament layers is explained by the direction of filament layers deposition, which is oriented parallel to the length of the stretcher with one component of the force acting perpendicular to this direction and tending to detach the layers.

From the research conducted we can say that the application of this technology in the furniture field is possible and beneficial with large perspective, especially for chairs with complex joints, involving more stretchers and various jointing angles. Three-dimensional printed connectors are also a benefit to product design, with the potential to combine colors and structures.

Further research work will be focused on changing the direction of filament deposition and adjusting the printing parameters, so to avoid the occurrence of major defects in 3D printing.

## Figures and Tables

**Figure 1 materials-15-02907-f001:**
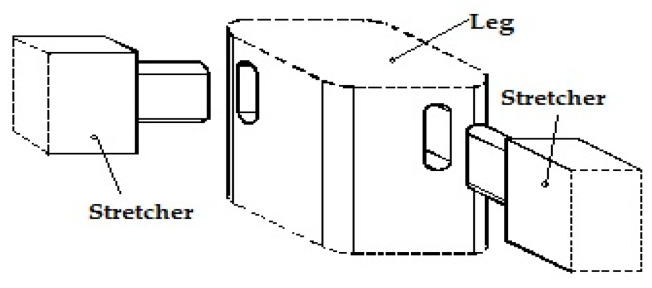
Mortise–tenon-jointed chair leg and stretchers.

**Figure 2 materials-15-02907-f002:**
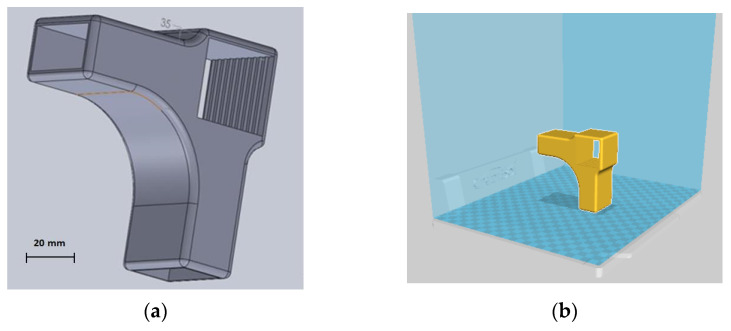
Designed connector for chair leg and stretchers: (**a**) 3D modelled part; (**b**) modelled part in the printer software prepared for 3D printing.

**Figure 3 materials-15-02907-f003:**
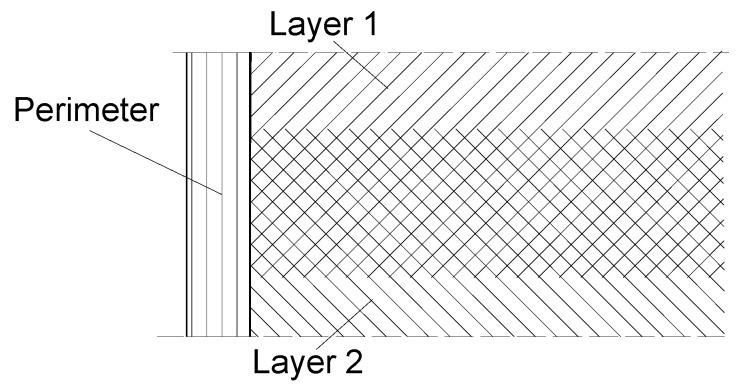
Perimeter and layer deposition of the filament.

**Figure 4 materials-15-02907-f004:**
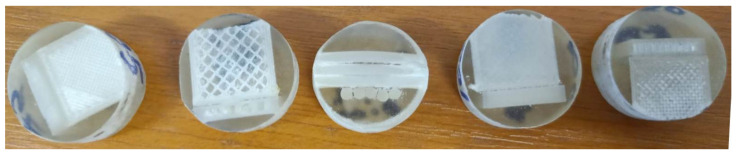
Filament and parts samples prepared for microscopic investigation.

**Figure 5 materials-15-02907-f005:**
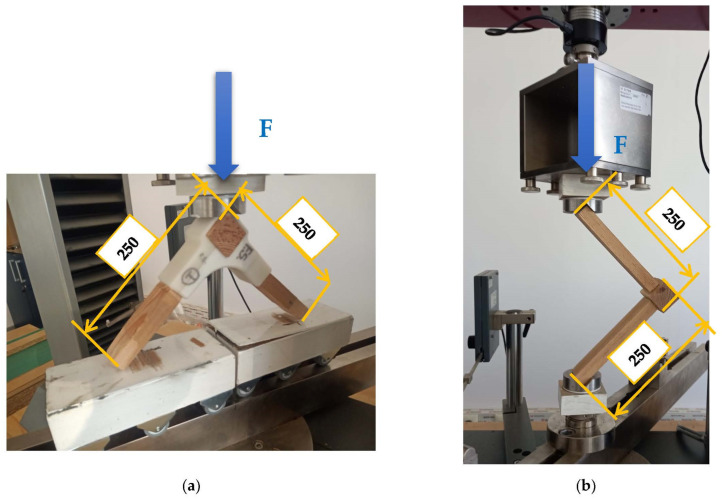
Corner-joint test specimens: (**a**) diagonal tensile testing; (**b**) diagonal compression.

**Figure 6 materials-15-02907-f006:**
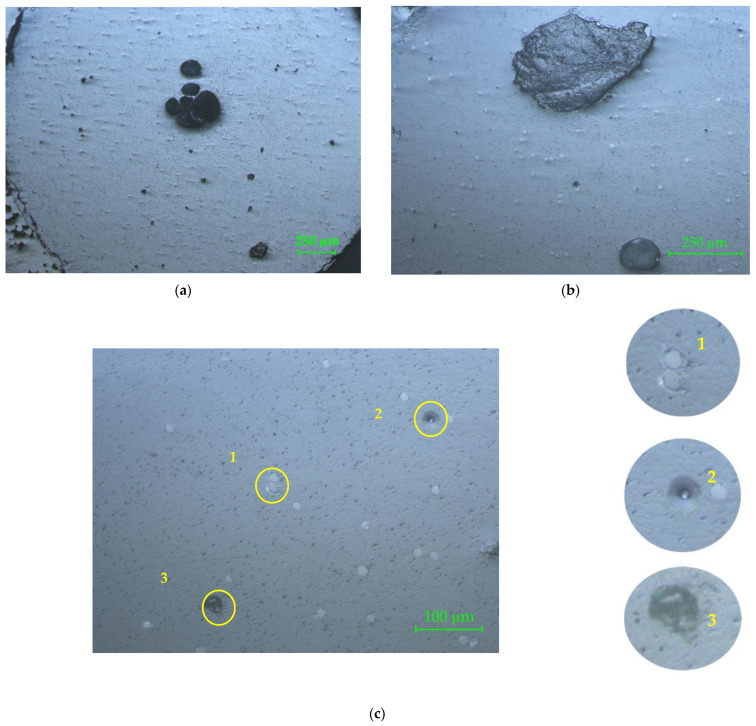
Microscopy of the cross-cut section of the PLA–glass filament: (**a**) 50×; (**b**) 100×; (**c**) 200×. The circled areas represent defects of the reinforcement material: the matrix did not adhere to the reinforcement (no. 1); gap left by the evulsion of a non-adherent glass fiber to the matrix (no. 2); common evulsion of the glass fiber in the matrix, occurred after cutting the sample (no. 3).

**Figure 7 materials-15-02907-f007:**
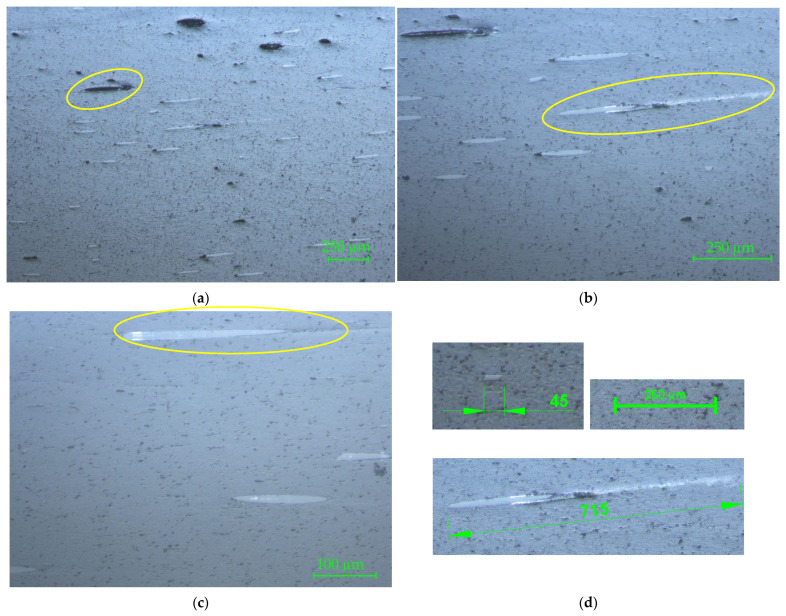
Microscopy of the longitudinal section of the PLA-glass filament: (**a**) 50×; (**b**) 100×; (**c**) 200×. The circled areas represent, as follows: (**a**) glass fiber disposed longitudinally in the right position; (**b**) the longest glass fiber measured on the sample; (**c**) partial exposure of the glass fibers; (**d**) measured lengths of the glass fibers (the smaller and the longest one).

**Figure 8 materials-15-02907-f008:**
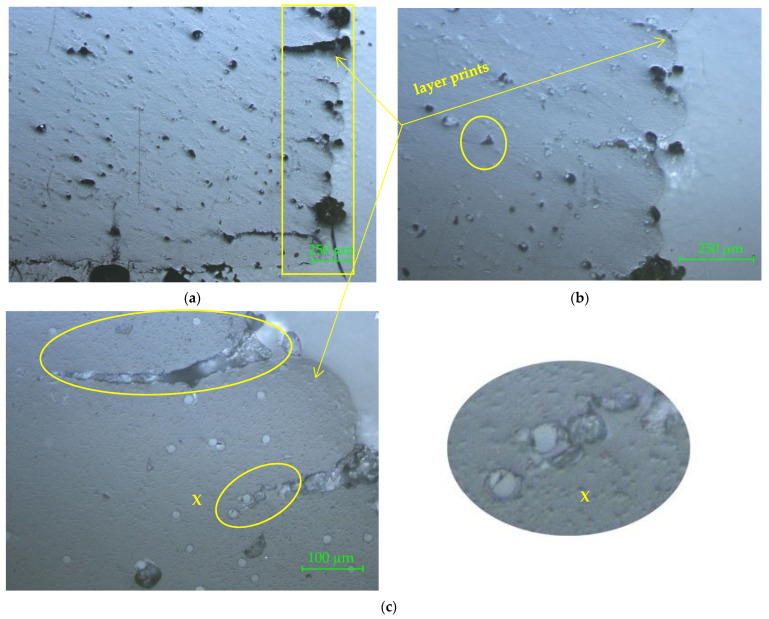
Connector 1 sample. Crosscut section: (**a**) 50×; (**b**) 100×; (**c**) 200×. The marked areas represent the defects which occurred on the 3D printed connectors: (**a**) prints of the deposited layers; (**b**) triangular defect specific to PLA; (**c**) delamination between layers and fiberglass bundle in detail X.

**Figure 9 materials-15-02907-f009:**
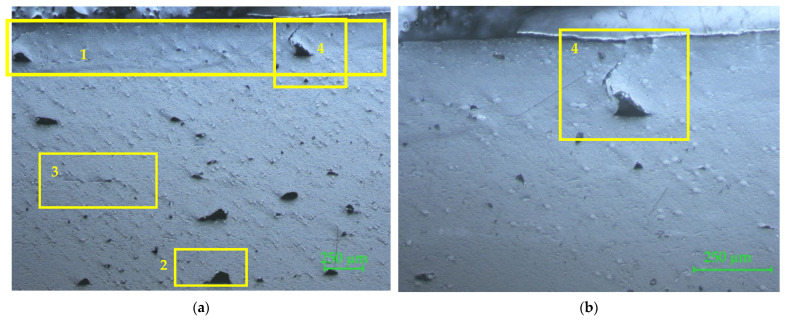
Connector 2 sample. Crosscut section: (**a**) 50×; (**b**) 100×. The marked areas represent, as follows: (**a**) perimeter marked with 1; overlaid triangular defects marked with 2, fiber–matrix unbound marked with 3, end of the perimeter contour marked with 4; (**b**) detail marked with 4 more clearly shows the changes of the deposition direction of the layer and the defect which occurred.

**Figure 10 materials-15-02907-f010:**
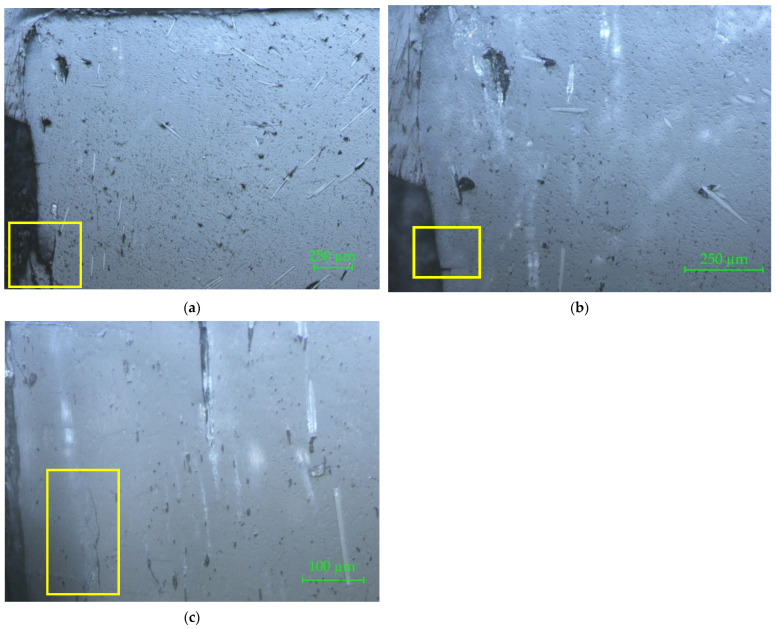
Connector 1 sample. Longitudinal section: (**a**) 50×; (**b**) 100×; (**c**) 200×. The marked areas represent, as follows: (**a**) delamination on the perimeter; (**b**) crack in the perimeter area; (**c**) crack between deposited layer and perimeter.

**Figure 11 materials-15-02907-f011:**
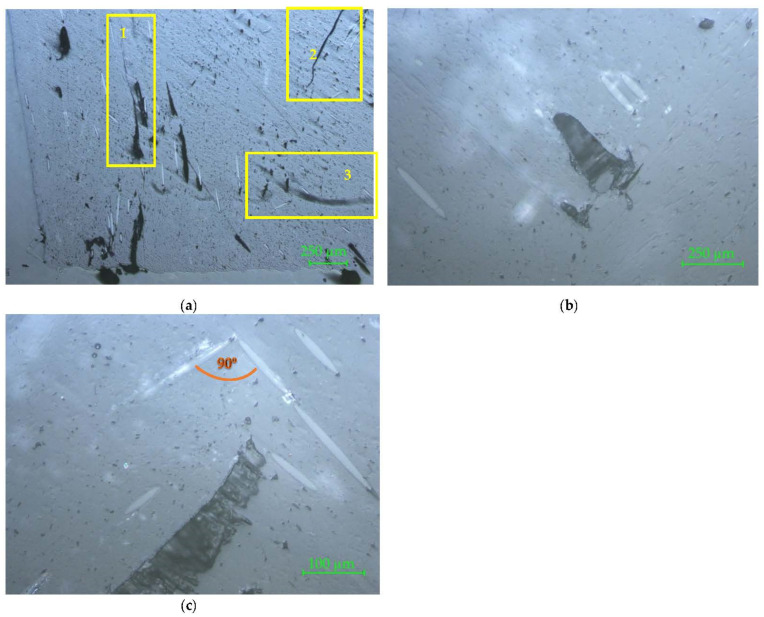
Connector 2 sample. Longitudinal section: (**a**) 50×; (**b**) 100×; (**c**) 200×. The marked areas represent, as follows: (**a**) areas marked with 1 and 2 show different types of cracks in the matrix, which can be considered macro-defects, and the zone marked 3 highlights the incomplete filling of the matrix.

**Figure 12 materials-15-02907-f012:**
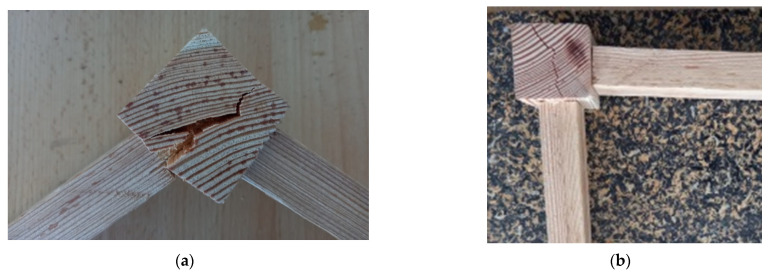
Larch wood corner joint after testing: (**a**) tensile; (**b**) compression.

**Figure 13 materials-15-02907-f013:**
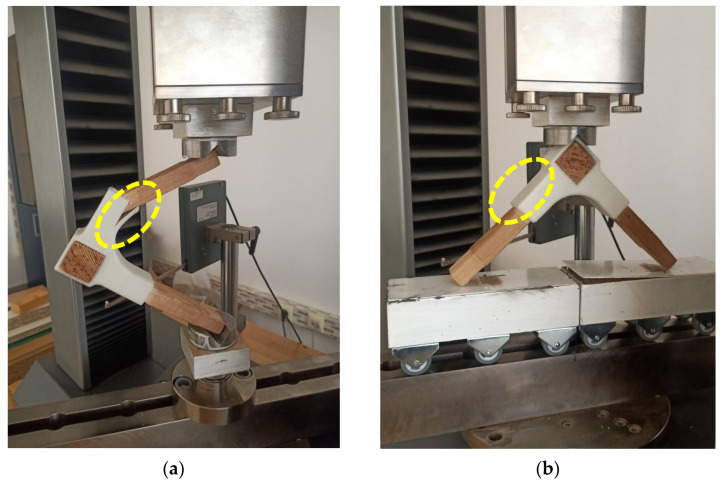
Tests of 3D printed connecting joint: (**a**) compression; (**b**) tensile.

**Figure 14 materials-15-02907-f014:**
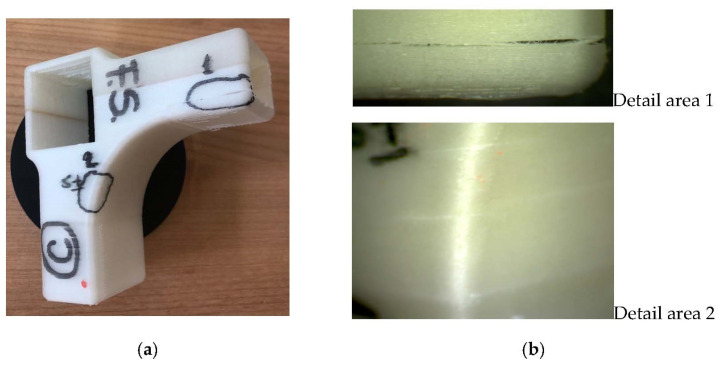
Three-dimensional printed connecting part after compression test: (**a**) the piece with marked areas of cracks; (**b**) details of cracks with 10× magnification.

**Figure 15 materials-15-02907-f015:**
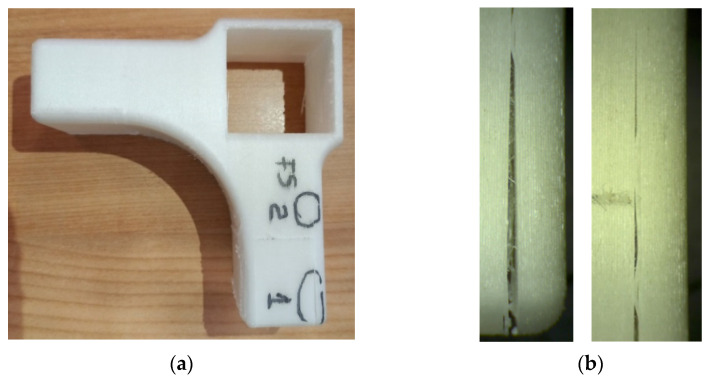
Three-dimensional printed connecting part after tensile test: (**a**) the piece with marked areas of cracks; (**b**) details of cracks with 10× magnification (left one for area 1 and right one for area 2).

**Table 1 materials-15-02907-t001:** Characteristics of PLA–glass composite filament [31].

Property	Value
Maximum tensile strength, MPA	57
Tensile strength at yield, MPa	46
Tensile modulus, GPa	4.0
Tensile elongation, %	3.4
Notched impact, J/m	29

**Table 2 materials-15-02907-t002:** Results of the mechanical tests.

Bending Moment, in Nm	Joint Type	
	Mortise–Tenon Joint	3D Printed Connecting Joint
*M_t_*	27.4 (1.6)	18.6 (1.9)
*M_c_*	28.2 (0.9)	12.1 (1.1) ^1^

^1^ Values in the parenthesis are standard deviations.

## Data Availability

Not applicable.

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
