# Peer review of "3D Printing Application in Wood Furniture Components Assembling"

_materials, 2022, doi:10.3390/ma15082907_

Round 1

Reviewer 1 Report

I read the paper 3D Printing Application in Wood Furniture Components Assembling reporting the work of a novel way to performwood-based components assembly. The paper is surely of interest in the field of wood furniture, but I would better highlight the advantages it can bring. Why FFF 3D printing could be beneficial in such a large scale fabrication?

In the following lines, my comments poit by point:

  1. The paper is well written and easy to understand. I think there is some missing literature, e.g. the paper Krzyżaniak, Ł.; Kuşkun, T.; Kasal, A.; Smardzewski, J. Analysis of the Internal Mounting Forces and Strength of Newly Designed Fastener to Joints Wood and Wood-Based Panels. Materials 2021, 14, 7119. https://doi.org/10.3390/ma14237119 is not cited. I think it perfectly fits in the context of your work.
  2. In figure 2 a scalebar is recommended to make the reader realize the dimensions you are talking about.
  3. The captions should perfectly describe the pictures without need to read the whole article. Please,improve (e.g. indicating what the circled elements mean).

Author Response

Answers to Reviewer 1

Comments and Suggestions for Authors

I read the paper 3D Printing Application in Wood Furniture Components Assembling reporting the work of a novel way to performwood-based components assembly. The paper is surely of interest in the field of wood furniture, but I would better highlight the advantages it can bring. Why FFF 3D printing could be beneficial in such a large scale fabrication?

Answer: The answer related to the benefit of using 3D printed is presented at page 2, lines 83-87: ”The present research has as objective to replace the basic glued mortise-tenon joints used for assembling the chair leg and stretchers with a 3D printed connector in which the three chair components are introduced. This replacement brings the advantage of easing the assemble process and reducing the production time, the operations of processing and gluing the tenon-mortise joints being not necessary this case”.

 and at the Conclusion chapter:

“From the research conducted we can say that the application of this technology in furniture field is possible and beneficial with large perspective, especially for chairs with complex joints, involving more stretchers and various jointing angles. 3D printed connectors are also a benefit in product design, by combining colors and structures.”

In the following lines, my comments poit by point:

  1. The paper is well written and easy to understand. I think there is some missing literature, e.g. the paper Krzyżaniak, Ł.; Kuşkun, T.; Kasal, A.; Smardzewski, J. Analysis of the Internal Mounting Forces and Strength of Newly Designed Fastener to Joints Wood and Wood-Based Panels. Materials 202114, 7119. https://doi.org/10.3390/ma14237119 is not cited. I think it perfectly fits in the context of your work.

Answer: The reference was added at line 82, page 2.

  1. In figure 2 a scalebar is recommended to make the reader realize the dimensions you are talking about.

Answer: The scalebar was added in Fig. 2a.

  1. The captions should perfectly describe the pictures without need to read the whole article. Please,improve (e.g. indicating what the circled elements mean).

Answer: The captions were improved, as suggested.

The authors would like to thank the Reviewer 1 for the remarks and recommendations.

Reviewer 2 Report

The manuscript presents the results from application of additive manufacturing in furniture manufacturing, namely the design and development of a 3D-printed furniture connector for chair components. Overall, the manuscript is well-written, structured and informative, and can be accepted for publication in Materials Journal after a major revision. Please, see below my comments on your work:

The abstract of the manuscript (lines 10 to 31) and the keywords (line 32) correspond to the title, aims and objectives of the paper. The abstract is informative, and contains the main findings of the article. Please add more specific results obtained from your research work in the abstract, now it is a bit general.

Line 19: please provide the full term, i.e. “polylactic acid”, and then the common abbreviation PLA.

Line 29: please delete “have”, it should be past simple tense.

In the keywords, I’d recommend to add also “polylactic acid (PLA)”.

Line 55: please add some sentences on the main characteristics and advantages of PLA as a functional material. Please check this relevant references:

https://doi.org/10.3390/polym14061227

https://doi.org/10.1089/3dp.2016.0054

Line 89: please avoid using commercial product names, e.g. FILATICUM PLA in scientific articles.

Line 122-123: please explain the selected 3D printing parameters.

In general, the Introduction part is well written and informative, and provides relevant information on the topic of the research, based on previously published studies.  It can be further extended based on the above given comments.

Line 130: please explain why did you choose European larch wood for manufacturing chair leg and stretchers?

Line 138: please add the standard ASTM D638 in the references of your paper.

Overall, the Materials and Methods section is very well written and detailed.

The Results and Discussion is informative, but is not properly discussed with previous research works in the field. The authors compared their results with only 2 articles. Please extend the discussion part of your manuscript.

The Conclusion part reflects the main findings of the research. Here I’d recommend to add how the results of your research work can be integrated into furniture manufacturing practice.

The references cited are appropriate and correspond to the topic of the manuscript.

Best regards!

Author Response

Answers to Reviewer 2

The manuscript presents the results from application of additive manufacturing in furniture manufacturing, namely the design and development of a 3D-printed furniture connector for chair components. Overall, the manuscript is well-written, structured and informative, and can be accepted for publication in Materials Journal after a major revision. Please, see below my comments on your work:

The abstract of the manuscript (lines 10 to 31) and the keywords (line 32) correspond to the title, aims and objectives of the paper. The abstract is informative, and contains the main findings of the article. Please add more specific results obtained from your research work in the abstract, now it is a bit general.

Answer: The abstract was improved by adding the undelined text below:

In line 23: For this purpose, several evaluation methods were applied: microscopic investigation with 50X, 100X and 200X magnifications, both of the filament and of the 3D printed connector....

In line 27: The microscopic investigation of the composite filament revealed the agglomerations of glass fibers into the core matrix and areas where the distribution of the reinforcements was poor. The heterogeneous structure of the filament and the defects highlighted in the 3D printed connectors by the microscopic investigation contributed to the mechanical behavior of L-type connecting joints.

Line 19: please provide the full term, i.e. “polylactic acid”, and then the common abbreviation PLA.

Answer: Done.

Line 29: please delete “have”, it should be past simple tense.

Answer: Done.

In the keywords, I’d recommend to add also “polylactic acid (PLA)”.

Answer: Done.

Line 55: please add some sentences on the main characteristics and advantages of PLA as a functional material. Please check this relevant references:

https://doi.org/10.3390/polym14061227

https://doi.org/10.1089/3dp.2016.0054

Answer: Done.

Line 89: please avoid using commercial product names, e.g. FILATICUM PLA in scientific articles.

Answer: The product name was deleted.

Line 122-123: please explain the selected 3D printing parameters.

Answer: The explanation was added at line 123, as follows:

”The 3D printing parameters were selected according to the recommendations made by the filament manufacturer and following the preliminary results obtained by the 3D printed parts in terms of mechanical strength for different printing parameters.”.

In general, the Introduction part is well written and informative, and provides relevant information on the topic of the research, based on previously published studies.  It can be further extended based on the above given comments.

Line 130: please explain why did you choose European larch wood for manufacturing chair leg and stretchers?

Answer: The explanation was added at line 130:

”Larch wood is a resinous wood with good stability and small moisture movement. It is appreciated for its fine structure, for its toughness and high density compared to other resinous woods. It is strong and sturdy and it is often used for yacht building and joinery, stairs and floors, as well as for external cladding projects and outdoor furniture, due to its good resistance to weathering. In furniture industry larch wood is mainly used for chairs, benches and tables and has a large availability on the market at low cost.”.

Line 138: please add the standard ASTM D638 in the references of your paper.

Answer: The reference was added [31].

Overall, the Materials and Methods section is very well written and detailed.

Answer: Thank you for your appreciation.

The Results and Discussion is informative, but is not properly discussed with previous research works in the field. The authors compared their results with only 2 articles. Please extend the discussion part of your manuscript.

Answer: The results are now compared with more articles, as follows:

Line 200: ”The PLA porosity was also noticed by SEM micrographs of PLA’s surface [16].”

Line 286 ” New designed 3D printed fasteners used for corner joints of beech wood and tested under tension and compression recorded lower bending moment values, namely 12.34 Nm for tensile and 8.57 Nm for compression tests [26].”

Line 296 ”3D printed connectors (with and without inner wall) used for the frontal parallel joints [19] and tested for bending load were fractured in the PLA material only for the variant with inner wall, because of the force acting perpendicular to the deposition layers and forcing the end of the connector to crack. The defects occurred on the connectors are similar to those presented in Figure 13a.”

The Conclusion part reflects the main findings of the research. Here I’d recommend to add how the results of your research work can be integrated into furniture manufacturing practice.

Answer: A phrase was added at the Conclusion chapter, as follows:

”From the research conducted we can say that the application of this technology in this furniture field is possible and beneficial with large perspective., especially for chairs with complex joints, involving more stretchers and various jointing angles. 3D printed connectors are also a benefit in product design, by combining colors and structures.”

The references cited are appropriate and correspond to the topic of the manuscript.

Answer: Thank you. The recommended references were also added.

Best regards!

The authors would like to thank the Reviewer 2 for the remarks and recommendations.

Round 2

Reviewer 2 Report

The authors have addressed all my previous comments/remarks. I believe the manuscript can be accepted for publication in its current form.